# *Euterpe oleracea* Mart (Açaizeiro) from the Brazilian Amazon: A Novel Font of Fungi for Lipase Production

**DOI:** 10.3390/microorganisms10122394

**Published:** 2022-12-02

**Authors:** Iracirema S. Sena, Adriana M. Ferreira, Victor H. Marinho, Fabrício H. e Holanda, Swanny F. Borges, Agerdanio A. de Souza, Rosemary de Carvalho R. Koga, Adilson L. Lima, Alexandro C. Florentino, Irlon M. Ferreira

**Affiliations:** 1Biocatalysis and Applied Organic Synthesis Group, Department of Exact Sciences, Federal University of Amapá, Josmar Chaves Pinto Highway, KM—02 Bairro Zerão, Macapá 68902-280, AP, Brazil; 2Pharmaco Research Laboratory, Federal University of Amapá, Josmar Chaves Pinto Highway, KM—02 Bairro Zerão, Macapá 68902-280, AP, Brazil; 3Biological Control Laboratory, Brazilian Agricultural Research Corporation (Embrapa), Josmar Chaves Pinto Highway, KM—05 Bairro Zerão, Macapá 68903-419, AP, Brazil

**Keywords:** Amazon fungi, hydrolases, açai, microorganisms

## Abstract

Lipases (EC 3.1.1.3) are hydrolases that catalyze triglycerides hydrolysis in free fatty acids and glycerol. Among the microorganisms that produce lipolytic enzymes, the entophytic fungi stand out. We evaluated 32 fungi of different genera, *Pestalotiopsis*, *Aspergillus*, *Trichoderma*, *Penicillium*, *Fusarium*, *Colletotrichum*, *Chaetomium*, *Mucor*, *Botryodiplodia*, *Xylaria*, *Curvularia*, *Neocosmospora* and *Verticillium*, isolated from *Euterpe oleracea* Mart. (Açaizeiro) from the Brazilian Amazon for lipase activity. The presence of lipase was evidenced by the deposition of calcium crystals. The endophytic *Pestalotiopsis* sp. (31) and *Aspergillus* sp. (24) with Pz 0.237 (++++) and 0.5 (++++), respectively, were the ones that showed the highest lipolytic activity in a solid medium. Lipase activity was rated in liquid medium, in a different range of temperatures (°C), pH and time (days). The values obtained in the production of lipase by the endophytic fungi were 94% for *Pestalotiopsis* sp. (31) and 93.87% for *Aspergillus* sp. (24). Therefore, it is emphasized that the endophytic fungus isolated the *E. oleracea* palm may be a potential candidate to produce enzymes of global commercial interest.

## 1. Introduction

Brazil has around 20% of the world’s biodiversity [1], and the Amazon is majorly responsible for this biodiversity, due to the high specificity of the environment, which contributes to a diversified species of microorganisms; however, the microorganisms present and their interactions with other organisms are poorly understood [2,3,4].

*Euterpe oleracea* Mart. (Arecaceae), commonly known as açaí, is a palm tree typically found in the Amazon region, naturally found in North Brazil, especially in the Pará, Amazonas and Amapá states, and which has great importance because of the economic value of its fruit pulp [5]. Numerous advances have been made in recent years to demonstrate the health benefits of açai pulp and seed from *E. oleracea* Mart [6]. Studies have confirmed it to be one of the most potent antioxidants [7] and anti-inflammatory food sources available, attributable to a class of flavones, as well as other polyphenols, lignans and saccharides [8].

Endophytic fungi are microorganisms that live inside the host plant tissues without causing diseases [2]. Some studies show that fungi endophytes are capable of producing a large number of important bioactive metabolites, for example, the taxol, an important anticancer drug, is produced by fungus endophytic *Taxomyces andreanse*, isolated from *Taxus brevifolia* bark [9]. The endophytic fungi can be also a source of different enzymes, such as, amylases, quietness, proteases, asparaginases, celluloses, laccase and lipases with biotechnological interest [10,11,12].

The first description of endophytic fungi from *E. oleracea* palm leaves, of the Amazon region, was performed by Rodrigues [13]. In opportunity, Rodrigues also described the occurrence of a new genus, such as, *Letendraeopsis palmarum*, and new species of the genus *Idrella* isolated from *E. oleracea* palm [14]. The enzymatic potential (cellulolytic and amylolytic) of endophytic fungi isolated from *Euterpe precatoria* Mart. was shown by Batista [15]; Recently, the extract with effect antimicrobial from endophytic fungi of *E. precatoria* against *Staphylococcus aureus*, *Streptococcus pneumoniae*, *Enterococcus faecalis*, *Escherichia coli* and *Klebsiella pneumoniae* human pathogens was examined [16]. In addition, endophytic fungi from *E. precatoria* were used as antagonistic agents towards *Colletotrichum gloeosporioides* in the control of anthracnose in açaí leaflets [17]. McCulloch et al. [18] reported the genome, in a single chromosome, of *Lactococcus lactis* strain AI06, isolated from the mesocarp of the açaí fruit (*Euterpe oleracea*) in eastern Amazonia, Brazil. This strain is an endophyte of the açaí palm and also a component of the microbiota of the edible food product. However, studies about endophytic fungi from *E. oleracea* are scare in the literature.

Lipases (EC 3.1.1.3) are catalytic enzyme of hydrolases reaction, on the carboxylic ester bond and catalyze the reaction of hydrolysis esterification and interesterifications of fats with excellent performance [19,20]. These enzymes correspond to the third biggest selling group in the world [21], your applications goes from the production of detergents, degreasers [22,23], textile products, to paper, [10,23]. Lipase can be produced by animals, plants and microorganisms, the enzymes produced by the last one being more stable when compared to other sources [19].

Some studies are being carried out to explore environments that have not been studied, especially located in the Amazon region. Therefore, this article aims to demonstrate the potential enzymatic (lipolytic) activity of endophytic fungi from the fruit of *Euterpe oleracea* Mart (açaizeiro).

## 2. Materials and Methods

### 2.1. Reagents and Solvents

4-Nitrophenyl palmitate (99%) was obtained from Sigma-Aldrich (São Paulo, Brazil). Isopropanol (98%) was obtained from Synth (São Paulo, Brazil). Agar and Malt extract were obtained from Kasvi (São Paulo, Brazil).

### 2.2. Isolation and Identification of Endophytic Fungi from the Fruits of E. oleracea Mart

The botanical material (fruits and leaves) of *E. oleracea* was collected and given by the Brazilian Agriculture and Livestock Research Company (Empresa Brasileira de Pesquisa Agropecuaria—EMBRAPA/Amapá), from the coordinates (N 00°22′ 55″ e O 51°01′40″) during the period of August 2018. The fungi used in this research were isolated from their fruits, roots and leaves and were stocked according to [24]. All microbiologic manipulation activity was conducted inside the laminar flow cabinets.

### 2.3. Identification and Conservation of Endophytic Fungi

The morphological identification was conducted at the genus level by macro-morphological grouping, by observing the characteristics of each individual such as its appearance, form, color and the consistency of its colonies. In order to visualize the microscopical structures, specially designed coverslips were made in microcultures, using a blue pigment of lactophenol (0.5%) across the surface of the coverslips. An optical microscope (OLYMPUS ^®^ BX41) was used for capturing these images. The images were amplified by 200–400× and compared to the ones found in the specialized literature [13]. Thereafter, the macromorphological and micromorphological analysis identified the following genera: *Pestalotiopsis* sp., *Aspergillus* sp., *Trichoderma* sp., *Penicillium* sp., *Fusarium* sp., *Colletotrichum* sp., *Chaetomium* sp., *Mucor* sp., *Botryodiplodia* sp., *Xylaria* sp., *Neocosmospora* sp. and *Verticillium* sp.

### 2.4. Determining the Enzymatic Activity

The microorganisms were precisely inoculated in the center of the Petri dish (90 cm) and incubated in B.O.D in a regulated environment of 28 °C with a photoperiod of 12 h. The measurement of the colony and the halo diameter were expressed in centimeters (cm) and conducted once per 24 h for 5 consecutive days. All the testing was conducted in triplicate.

### 2.5. Enzymatic Test for Lipase in a Solid Environment

For its lipolytic activity, the fungi were cultivated in Peptone Agar (peptone, 8 g; sodium chloride, 4 g; calcium chloride, 0.08 g; agar, 16 g; distilled water, 800 mL and pH-6.0) Tween 20 was sterilized separately and supplemented at 1% (*v/v*) in the growth medium. The enzymatic reaction was considered a success once the formation of calcium salt crystals and lauric acid produced by the enzyme was confirmed. Lipolytic activity, the value of Pz (zone of precipitation), was calculated by using the following equation:$$Pz=\frac{Colony\;diameter}{Colony\;diameter+Zone\;of\;precipitation}$$

The *Pz* value of the lipolytic activity was classified into 5 categories: *Pz* = 1 negative lipasic activity (-); *Pz* < 0.90–0.99 weak lipasic activity (+); *Pz* < 0.80–0.89 = poor lipasic activity (++); *Pz* < 0.70–0.79 = moderate lipasic activity (+++); and *Pz* < 0.70 = intense lipasic activity (++++).

### 2.6. Lipase Production in a Liquid Environment

After the sorting with the ones that were isolated in a solid medium, we opted to optimize the lipase production in a liquid environment using only the fungi *Pestalotiopsis* sp. (31) and *Aspergillus* sp. (24). In order to cultivate those in a liquid medium, Erlenmeyer flasks (250 mL) were used, containing 25 mL of growth medium with 5 mycelial discs (8 mm of diameter) of the fungi *Pestalotiopsis* sp. (31) and *Aspergillus* sp. (24). After cultivating them, those fungi were incubated for different days (3, 6 and 9 days) under different temperatures (25, 30 and 35 °C) and different pH (5, 7 and 9) in an orbital shaker at 150 rpm. The fermented broth was vacuum-filtered with a filter paper of 80 g.m^−2^. Its biomass was vacuum-drained until its weight reached a constant weight in an average of about 96 h. The filtered broth devoid of cells was used to determine the lipolytic activity; for this end, olive oil was used as the carbon font and inductor of the lipase production. the medium used possessed the following composition: 37g/L of peptone, 1.11 g/L of magnesium sulfate, 1.85 g/L of potassium phosphate, 1.85 g/L of sodium nitrate and 14 mL/L of olive oil. The experiments were realized in triplicate.

### 2.7. Quantification of Lipolytic Activity in a Liquid Medium

The lipolytic activity was measured according to the methodology described and adapted by Mayordomo et al. [25]. The lipolytic activity was conducted using 250 µL of a solution containing 200 mg Triton X-100, 50 mg of gum arabic and phosphate buffer (0.1 M) at pH = 7.5 for a final volume of 50 mL. Following this, 250 µL of the enzymatic broth was added to a 45 µL solution of palmitate of *p*-nitrophenol (*p*-NPP) diluted in isopropanol (10 mL); thereafter, the reaction was taken to a water bath at 40 °C at 30 min. Then, 0.5 mL of Trizma base 2% (*m*/*v*) was added. The quantification of the lipolytic activity was realized starting with the solution *p-*NPP that releases nitrophenol (NP), quantified by absorbance at 398 nm in a spectrophotometer (Perkinelmer-Lambda35). One unit of activity was defined as the amount of enzyme required to hydrolyze 1 µmol of *p-*NPP per minute under the conditions described.

### 2.8. Statistical Analysis

#### Experimental Design and Statistical Model

In this study, a three-level and three-variable *Box–Behnken* factorial design was applied to determine the best combination of variables for determination of lipase production using isolated endophytic fungi. The pH of the medium, time (days) and temperature (°C), which were identified to have strong effects on the response in preliminary one-factor-at-a-time experiments, were taken as the variables tested in a 15-run experiment to determine their optimum levels. Independent variables were designated as *x*1, *x*2 and *x*3, and their level values are shown in Table 1. The polynomial equation used for the three variables is given below:Y = β0 + β1 × 1 + β2 × 2 + β3 × 3 + β11 × 1^2^ + β22 × 2^2^ + β33 × 3^2^ + β12 × 1 × 2 + β13 × 1 × 3 + β23 × 2 × 3
where Y is the predicted response; β0 is model constant; β1, β2 and β3 are the linear coefficients; β11, β22 and β33 are the quadratic coefficients; β12, β13 and β23 are the interaction coefficients; and *x*1, *x*2 and *x*3 are independent variables.

The optimal condition was determined considering the lipase production content (BD%) as the response. The software STATISTICA^®^ (version 10, Statesoft—Inc., Tulsa, OK, USA, trial version, 2011) was used for experimental design, data analysis and determination of optimal conditions. ANOVA was used for the evaluation of the significance of independent variables’ influence and interactions. Pareto charts were applied to obtain the significance of the impact of tested variables on mentioned responses.

## 3. Results and Discussion

### 3.1. Isolating and Purifying Endophytic Fungi

The strains of endophytic fungi were isolated from *E*. *oleracea* by making use of fragments of its plant tissue (fruits, roots and leaves), Figure 1. A total of 32 fungi of different morphological genera were isolated, including: *Pestalotiopsis* sp., *Aspergillus* sp., *Trichoderma* sp., *Penicillium* sp., *Fusarium* sp., *Colletotrichum* sp., *Chaetomium* sp., *Mucor* sp., *Botryodiplodia* sp., *Xylaria* sp., *Curvularia* sp., *Neocosmospora* sp. and *Verticillium* sp. Figure 2 shows the endophytic fungi isolated and grown in Petri dishes after the process of purifying their lineages.

Rodrigues [14] isolated endophytic fungi from *Euterpe oleracea*, obtaining 21–30% isolation rate, and Southee and Johnson [26] reported an isolation frequency of 20.3% in two species of palm (*Sabal bermudana* and *Livistona chinensis*).

The taxon *Pestalotiopsis* is characterized by spores with pigmented median cells, divided by four eusepta (true septum), with 2–3 apical appendages resulting from tubular extensions of the apical cell and a central basal appendage [27]. However, the genus *Pestalotiopsis* is complex and can be difficult to classify at the species level, because characteristics such as fruiting structure, length and conidia morphology tend to vary within species and also with any change in the environment [28]. The colonies were characterized by having white coloration, vigorousness, cottony mycelium, formation of black masses of conidia and abundant sporulation.

Rodrigues [14] recorded the first occurrence of the genus *Pestalotiopsis* in the Amazon region as endophytic in açaí leaves.

Additionally, the identification of *Aspergillus* has traditionally been based on morphological characterization [29]. Macromorphological characteristics include colony color on various culture media, colony diameter, colony reverse color, production of exudates and soluble pigments. The micromorphological characterization mainly related to the form of serialization of the conidial head, the size of the vesicle, the morphology of conidia and the presence of cells [30].

The *Aspergillus* conidiophore is simple, usually aseptic and ends in a vesicle, where the phialides are inserted. Some species can produce Hülle cells or sclerotia. Many species of *Aspergillus* have teleomorphs and reproduce sexually [30].

### 3.2. Screening of Lipase-Producing Endophytic Fungi

The determinant factors that make the qualitative enzymatic test selection viable include the direct correlation between the halo size and the degradative capacity of the microorganisms. The Table 2 show the enzymatic activity of the isolated endophytic fungi from endophytic *E*. *oleracea* (açaizeiro). The results from this selection allow to foresee the enzyme production yields, indicating the presence of a determined substance through the detection of some specific activity [31]. The isolated fungi with the largest activity halo were selected for the enzymatic activity determination step in a liquid environment.

From the alignment of lipase production of endophytic fungi in Petri dishes, the isolated strains show the formation of calcium crystal halos around the colonies (Figure 3), indicating the production of lipase by these fungi.

Based on the enzymatic determination index (Pz) obtained, it has been shown that not every endophytic fungus that were analyzed in this experiment exhibited lipolytic activity; out of the 32 fungi species, 11 have shown a strong enzymatic activity. Among these prominent species are: *Botryodiplodia* sp. (30) with a Pz value of 0.237 (++++), *Aspergillus* sp. (21) with a Pz value of 0.5 (++++), *Aspergillus* sp. (24) with a Pz value of 0.5 (++++), *Pestalotiopsis* sp. (31) with a Pz value of 0.4 (++++), *Neocosmospora* sp. (9) with a Pz value of 0.5 (++++), *Fusarium* sp. (2) with a Pz value of 0.5 (++++), *Fusarium* sp. (27) with a Pz value of 0.5 (++++), *Trichoderma* sp. (17) with a Pz value of 0.5 (++++), *Penicillium* sp. (5) with a Pz value of 0.7 (+++) moderate, *Penicillium* sp. (7) with a Pz value of 0.5 (++++), *Colletotrichum* sp. (3) with a Pz value of 0.5 (++++) and *Chaetomium* sp. (35) with a Pz value of 1, therefore without activity. The remaining 21 species were not able to be grown into a medium for lipolytic activity detection.

Most of the lipase-producing fungi are isolated from industrial or domestic oily leftovers, which have been contaminated with grease and oil and living and dead animals [32,33]. From this study, it is also possible to affirm that endophytic fungi can exhibit an interesting lipolytic activity. The lipasic activity is frequently measured by the release of either fatty acids or glycerol, and the use of a solid medium with inducting substrates such as vegetable oil, standard triglycerides, Tween 80 and coloring agents was already described in the literature, aiming at the pre-selection of lipase-producing microorganisms [34]. Many of these genera fungi were described in the literature as potential lipase producers, such as *Trichosporon*, *Botrytis*, *Pichia*, *Fusarium*, *Aspergillus*, *Mucor*, *Rhizopus*, *Penicillium*, *Geotrichum*, *Tulopsis* and *Candida* [35].

Tarci et al. [30] isolated *Aspergillus* sp. DPUA 1727 from both the maize and soil and it studied in order to produce lipase using agro-industrial waste as an inductor. It was shown that agro-industrial waste can be used for this purpose mainly if it presents a higher percentage of fatty acid esters (>80%).

### 3.3. Experimental Design for Lipase with Endophytic Fungi Pestalotiopsis sp. (30) and Aspergillus sp. (24)

The lipase production by the endophytic fungi *Pestalotiopsis* sp. (30) and *Aspergillus* sp. (21) from different experimental assays of the experimental planning protocol [32] concluded that the factorial planning Box–Behnken Design (BBD) might be the ideal tool in order to optimize the experimental conditions for endophytic fungi. Moreover, another significant advantage of using BDD instead of other techniques is the budget, due to the fact that it demands a smaller number of experimental executions and less time and, consequently, a smaller use of supplies [36].

The values of the variables, the levels used in the experiments and the results obtained are shown in Table 3. The variation between the maximum and minimum values obtained was from 93.18 to 94% for *Pestalotiopsis* sp. (31) and from 93.12 to 93.87% for *Aspergillus* sp. (24), where the highest percentages represent the higher relative production of lipase by the endophytic fungi compared to the negative control group, with the best responses (greater amount of lipase) in assay 4 for *Pestalotiopsis* sp. (31) and *Aspergillus* sp. (24) (pH 5; temperature of 35 °C; and time of 6 days). The coefficients of determination of the models (R^2^) were 0.82 for the tests of the endophytic fungus *Pestalotiopsis* sp. and 0.86 for the *Aspergillus* sp. (24). Model proficiency is demonstrated if R^2^ is equal to 0.75 or greater than this value [37]. The values obtained for the relative production of lipase by the endophytic fungi were quantified from the equation Y = 0.004489*X + 0.1323 generated by the standard curve (Figure 4).

Pareto charts of the standardized effects were generated that reveal the significant effects of the medium pH, growth time (days) and temperature (°C), both linear and quadratic, with the endophytic fungi *Pestalotiopsis* sp. (31) and *Aspergillus* sp. (24), where the bar length represents the absolute importance of the effects estimated according to the values used in the tests. The vertical line represents the boundary between significant and insignificant effects with a 5% risk of error. The effects are significant at a 95% confidence level in the experimental domain studied (*p* < 0.05) as shown in Figure 5.

As shown in Figure 5, three effects were statistically significant (p < 0.05) for relative lipase production with the endophytic fungus *Pestalotiopsis* (31) sp. For lipase activity from the endophytic *Aspergillus* sp. (24), evidently, the temperature variable (L) represents the most decisive factor for improving the production of lipase, and in these cases, as the values generated by the Pareto graph were positive (4.950 and 6.025), they show that the higher the temperature used in the reactions, the better. For the reaction conditions for the fungi, results can be observed in experiments 4, 8 and 12 of the experimental design matrix (Table 3).

The quadratic model correlating the factors utilized in the runs for lipase production by endophytic fungi allows the projection of three response surface graphics for the optimization of the results. The response surface graphic for the lipase production of the endophytic fungi *Pestalotiopsis* sp. (31) (Figure 6) was generated to determine the crossing between two conditions and analyze its effects over the obtained results.

The lipase production in response to the temperature and pH in the reactional medium (Figure 6A) reveals that a better production of this enzyme can be obtained due to the increase in temperature (from 30 to 35 °C) combined with both pH 5.0 and pH 9.0. When crossing the growth time and the pH of the medium (Figure 6B), it is possible to observe that the increase in the time interval used generates a good influence on the responses, increasing from 3 to 9 days with pH 5.0 and 9.0. In Figure 6C, the interaction between temperature and growth time shows that increases in both time and temperature are important in the results of relative lipase production.

Likewise, three response surface plots were obtained to optimize the experimental conditions for the endophytic fungus *Aspergillus* sp. (24) (Figure 7). Lipase production in feed at the temperature used and the pH of the rational medium (Figure 7A) show that the increase in temperature (>30 °C) combined with pH 5 and 9 directly influences the response. In Figure 7B, the relationship between time, pH and reaction time shows that, for a greater production of lipase, it is necessary to increase the reaction days (>6.0) using pH 5 or 9. In the interaction between reaction and temperature (Figure 7C), the best conditions obtained were at temperatures of 30 and 35 °C, mainly in the growth time of 9 days. It is important to highlight that the time and pH factors in the variables used in this experimental design were not statistically significant, making it evident that the temperature was the most relevant factor in this method.

Lipases are known for being efficient and stable catalyzers in many culture mediums and for acting in a diverse range of organic solvents. Many studies identified the production of enzymes by endophytic fungi in solids and liquids, just like the uses of those in transesterification reactions of many different lipidic biomasses [2,22,38].

In a study realized by Souza et al. 2018 [39] utilizing cotton oil as a substract, the *Preussia africana* isolated from *Handroanthus impetiginosus* showed lipolytic activity with 5.9 U/mL. *Stemphylium lycopersicie* isolated from *Humiria balsamifera* and *Sordaria* sp. isolated from *Tocoyena bullata* not only showed maximum activity for lipase production with 110 U/mL, but they also promoted the esterification reaction for the synthesis of ethyl oleate [2].

Between the factors that influenced the enzymatic activity, the medium’s pH is among the most significant variables in this study. The pH has a vital role in manutention of metabolism of fungi, taking part in a diverse range of biological functions. In the enzymatic process, each enzyme shows maximum activity at specific pH values. Moreover, the applicability of those enzymes in the biotechnological field depends on its stability in different pH ranges. In this sense, several studies report that endophytic fungi are efficient producers of stable enzymes with variable pH (alkaline and acid), such as lipases.

The endophytic *Aspergillus sojae* isolated from the plant *Plectranthus amboinicus* produced stable lipases, with maximum activity at pH 6 under a temperature of 27 °C [40]. Rocha et al. (2020) [2] isolated the fungi *Stemphylium lycopersici* from the leaves of a *Humiria balsamifera* and noted that under a temperature of 30° C and pH 7 it was a great lipase producer. According to [41], the fungi *Preussia africana* isolated from *Handroanthus impetiginosusi*, when cultivated under the conditions of pH 7 and a temperature of 37 °C, was an excellent lipase producer. Additionally, [42] optimized the lipase production process by making use of the fungi *Aspergillus niger* (MTCC 872) and observed that the maximum production of this enzyme was correlated to the temperature of 40 °C and pH 6.

The optimization of reaction conditions through experimental design has been a great tool for bioassays with fungi, as conducted in the production of lipase with three selected strains: *Candida guilliermondii*, *Penicillium sumatrese* and *Aspergillus fumigatus*. Enzymatic active were optimized through the experimental design, with excellent yield [43,44], by the endophytic fungus *Penicillium bilaiae*. Additionally, [45] applied a factorial experiment to optimize the production of extracellular enzymes by the endophytic fungus *Alternaria alternata.*

## 4. Conclusions

This study isolated 32 endophytic fungi from *Euterpe oleracea* (fruits, leaves and roots) from different genera. The isolated fungi of the endophytic *Pestalotiopsis* sp. (31) and *Aspergillus* sp. (24) were the ones that showed the highest lipolytic activity in a solid medium. Through experimental planning, the isolated *Aspergillus* sp. (24) and *Pestalotiopsis* sp. (31) have shown how the variable of pH affects the lipolytic activity, with the medium containing pH 9 having the most significance. Therefore, this highlights that the endophytic fungi isolated from the palm tree *E. oleracea* might be potential candidates for enzyme production of global commercial interest.

## Figures and Tables

**Figure 1 microorganisms-10-02394-f001:**
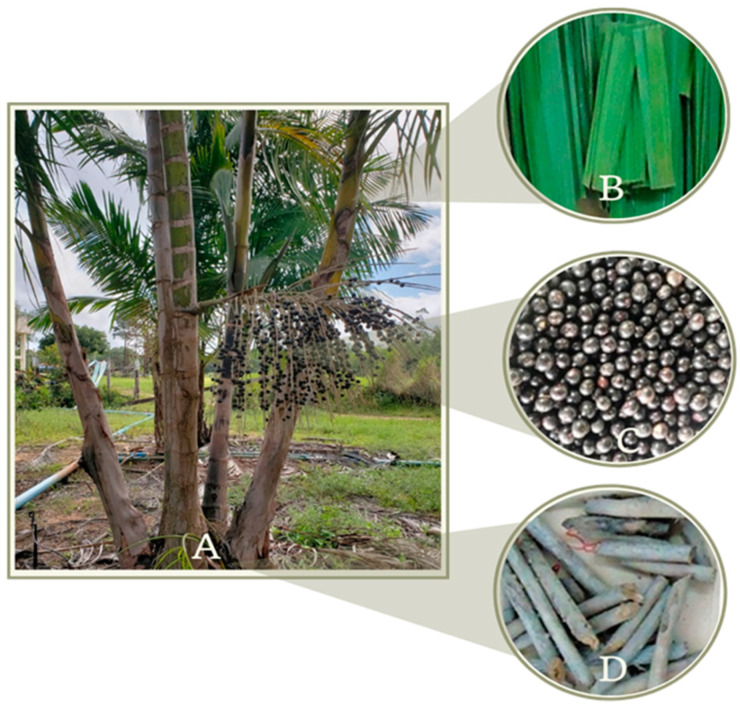
*E. oleracea* palm tree (**A**) leaves, (**B**) fruits and (**C**) roots (**D**).

**Figure 2 microorganisms-10-02394-f002:**
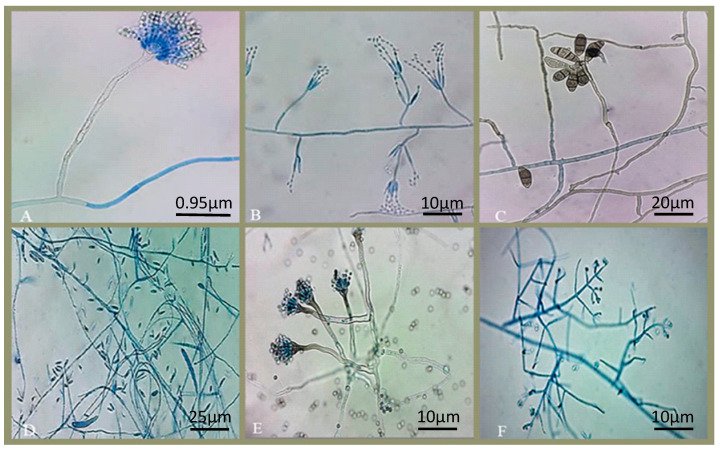
Fungi isolated from *E. oleracea* in solid medium morphological identification at the genus level in microculture observed by optical microscopy: (**A**) *Aspergillus* sp., (**B**) *Penicillium* sp., (**C**) *Pestalotiopsis* sp., (**D**) *Fusarium* sp., (**E**) *Aspergillus* sp. And (**F**) *Trichoderma* sp.

**Figure 3 microorganisms-10-02394-f003:**
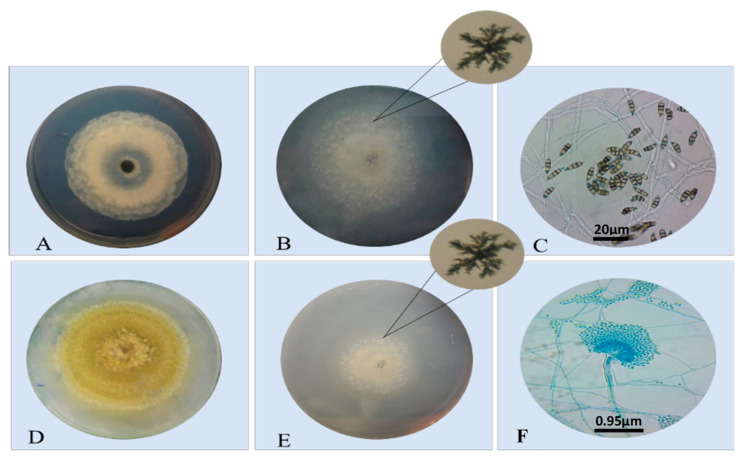
Mycelial growth of a *Pestalotiopsis* sp. (31) (**A**) and *Aspergillus* sp. (24) (**D**); semiquantitative test of enzyme production and the formation of crystal halos near the colony (**B**,**E**); morphology (**C**,**F**).

**Figure 4 microorganisms-10-02394-f004:**
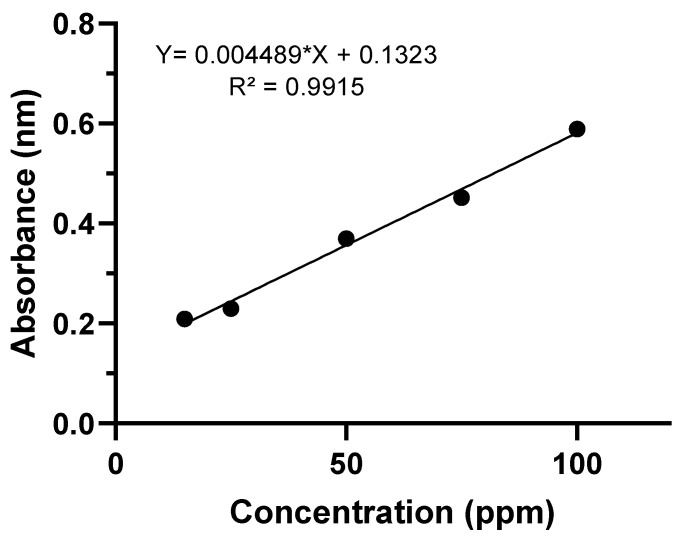
Standard curve used in the enzymatic determination lipolytic.

**Figure 5 microorganisms-10-02394-f005:**
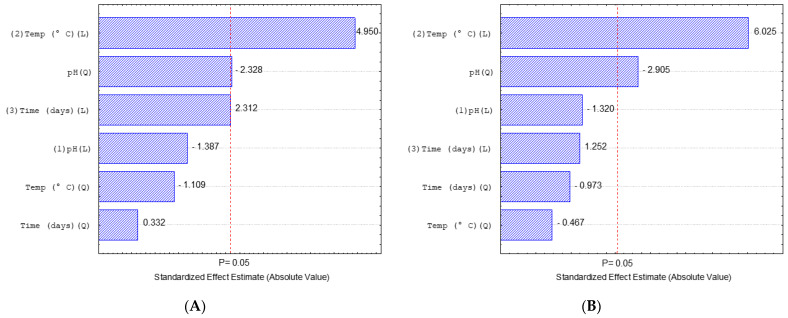
Pareto chart of effects for relative lipase production (%). (**A**) The endophytic fungus *Pestalotiopsis* sp. (31). (**B**) Endophytic fungus *Aspergillus* sp. (24).

**Figure 6 microorganisms-10-02394-f006:**
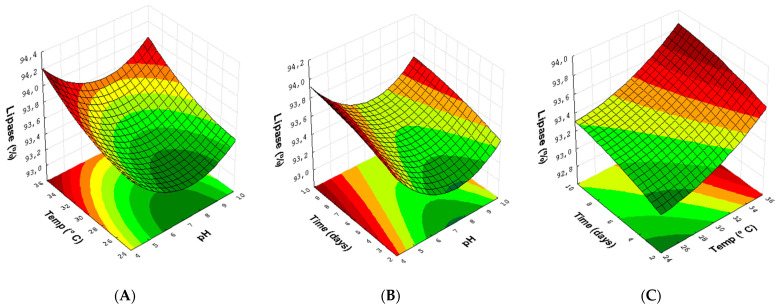
Response surface plots (3D) with the crossing of experimental conditions with the fungus *Pestalotiopsis* sp. (30): (**A**) the temperature (°C) and pH of the medium; (**B**) reaction growth time (days) and pH of the medium; and (**C**) temperature (°C) and reaction time (days).

**Figure 7 microorganisms-10-02394-f007:**
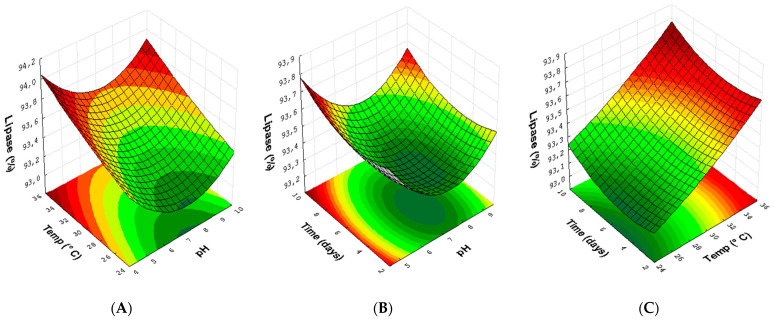
Response surface plots (3D) with the crossing of experimental conditions with the fungus. *Aspergillus* sp. (24): (**A**) the temperature (°C) and pH of the medium; (**B**) growth time (days) and pH of the medium; and (**C**) temperature (°C) and growth time (days).

**Table 1 microorganisms-10-02394-t001:** Three independent variables used in Box–Behnken factorial design.

Factor	Name	Levels		
		−1	0	+1
*x*1_1_	pH	5	7	9
*x*2_2_	Time (days)	3	6	9
*x*3_3_	Temperature (°C)	25	30	35

**Table 2 microorganisms-10-02394-t002:** Enzymatic activity of the isolated endophytic fungi from *E*. *oleracea* (açaizeiro), based on Pz calculations.

N°	Genus (Codig)	PZ *	N°	Genus (Codig)	PZ
1	*Botryodiplodia* sp. (30)	++++	17	*Aspergillus* sp. (26)	-
2	*Fusarium* sp. (2)	++++	18	*Aspergillus* sp. (11)	-
3	*Penicillium* sp. (5)	+++	19	*Penicillium* sp. (12)	-
4	*Penicillium* sp. (7)	++++	20	*Fusarium* sp. (13)	-
5	*Neocosmopora* sp. (9)	++++	21	*Mucor* sp. (14)	-
6	*Aspergillus* sp. (21)	++++	22	*Fusarium* sp(15)	-
7	*Aspergillus* sp. (24)	++++	23	*Penicillum* sp. (16)	-
8	*Fusarium* sp. (27)	++++	24	*Trichoderma* sp. (17)	++++
9	*Colletotrichum* sp. (3)	++++	25	*Verticillium* sp. (18)	-
10	*Pestalotiopsis* sp. (31)	++++	26	*Penicillium* sp. (19)	-
11	*Chaetomium* sp. (35)	-	27	*Penicillium* sp. (20)	-
12	*Aspergillus* sp. (1)	-	28	*Penicillium* sp. (22)	-
13	*Xylaria* sp. (4)	-	29	*Penicillium* sp. (25)	-
14	*Penicillium* sp. (8)	-	30	*Penicillium* sp. (25)	-
15	*Penicillium* sp. (10)	-	31	*Aspergillus* sp. (29)	-
16	*Aspergillus* sp. (23)	-	32	*Fusarium* sp. (34)	-

* PZ = Zone of precipitation.

**Table 3 microorganisms-10-02394-t003:** Experimental design matrix and responses to levels and variables.

Run	Coded and Uncoded Levels and Variables						Lipase (%) *Pestalotiopsis* sp. (30)	Lipase (%) *Aspergilllus* sp. (21)
	*x*1 (pH)		*x*2 (Temp.)		*x*3 (Time)			
1	5	−1	25	−1	6	0	93.40	93.23
2	5	−1	30	0	3	−1	93.23	93.66
3	5	−1	30	0	9	1	93.84	93.49
4	5	−1	35	1	6	0	94.00	93.87
5	7	0	25	−1	3	−1	93.18	93.12
6	7	0	25	−1	9	1	93.22	93.28
7	7	0	35	1	3	−1	93.61	93.50
8	7	0	35	1	9	1	93.68	93.70
9	9	1	25	−1	6	0	93.34	93.29
10	9	1	30	0	3	−1	93.41	93.40
11	9	1	30	0	9	1	93.54	93.56
12	9	1	35	1	6	0	93.67	93.61
13	7	0	30	0	6	0	93.36	93.40
14	7	0	30	0	6	0	93.37	93.31
15	7	0	30	0	6	0	92.38	93.24

## Data Availability

All data is provided in full in the results section of this paper.

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
