# Peer review of "Euterpe oleracea* Mart (Açaizeiro) from the Brazilian Amazon: A Novel Font of Fungi for Lipase Production"

_microorganisms, 2022, doi:10.3390/microorganisms10122394_

Round 1

Reviewer 1 Report

The subject of the manuscript Determination of lipase enzymatic activity by endophytic fungi isolated from Euterpe oleracea Mart (Açaizeiro) from the Brazilian Amazon is novelty and very interesting for readers.

The authors focused on lipases – enzymes (hydrolases) that catalyze triglycerides hydrolysis in free fatty acids and glycerol.

So. the screening of endophytic fungi of E. oleracea reveled the genera Pestalotiopsis sp, Neocosmospora sp, Colletotrichum sp, Fusarium sp, Penicillium sp, Aspergillus sp, Botryosphaeria sp., with similar lipase production profiles.

The presence of lipase was evidenced by the deposition of calcium crystals.

The authors concluded that the endophytic fungus isolated the E. oleracea palm may be potential candidates for the production of enzymes of global commercial interest. 

The text is clear and easy to read.

The manuscript is well written and has a significant contribution to the field.

The design research is well described.

The results are clearly presented.

The literature consulted is varied and relatively recently.

I have one observation to made: at the reference number 39, the name of species Phoenix Dactylifera L. is not italic. Please to correct!

Author Response

Reviewer 1:

Pont 1. I have one observation to made: at the reference number 39, the name of species Phoenix Dactylifera L. is not italic. Please to correct!

R = The text was corrected.

Pont 2. The manuscript contains a sufficient number of citations, although some of them are already outdated and I am missing some newer ones. The statistical analysis of the data is well done, but I have concerns about the input data, which should have been obtained on the basis of molecular identification of fungi and not just morphological, moreover, based on 60-year-old literature. This only morphological determination casts doubt on the authors' results, but can be easily remedied by additional species determination. 

L41: “However, studies about endophytic fungi from E. oleracea are limited in literature or scare”. They are not so rare, it is possible to mention these working with E. oleracea or a relative E. precatoria:

R = Refers all suggested were added in text.

L83-L84: This is the biggest shortage of work. The determination of the fungi should have been done on the basis of molecular methods that are already available today, and not on the basis of literature that is at least twenty years old (two sources are even from 1959 and 1971). In the list of "species" in Table 2, you include e.g. Aspergillus sp 26. It is clear that it does not have to be a specific species of Aspergillus - on the one hand, it does not have to be Aspergillus at all, on the other hand, it can be a strain, not the species. Individual strains may be of interest to other scientists, and if they isolate them from their sample and determine them molecularly (as is the standard today), it would be a benefit for science to be able to associate your ecological knowledge with this strain, which will not be possible in the current form of your results. What you labeled "Aspergillus sp 26" will be difficult for other scientists to replicate.

R = Concord as a reviewer on the importance of molecular identification to confirm genus and species. However, for this work it was not possible, however, for future work, some strains will certainly be treated for their species.

Reviewer 2 Report

The manuscript contains a sufficient number of citations, although some of them are already outdated and I am missing some newer ones. The statistical analysis of the data is well done, but I have concerns about the input data, which should have been obtained on the basis of molecular identification of fungi and not just morphological, moreover, based on 60-year-old literature. This only morphological determination casts doubt on the authors' results, but can be easily remedied by additional species determination. 

The most significant criticisms

L41: „However, studies about endophytic fungi from E. oleracea are limited in literature or scare.“ They are not so rare, it is possible to mention these working with E. oleracea or a relative E. precatoria:

McCulloch, J. A., de Oliveira, V. M., de Almeida Pina, A. V., Pérez-Chaparro, P. J., de Almeida, L. M., de Vasconcelos, J. M., ... & Nunes, M. R. T. (2014). Complete genome sequence of Lactococcus lactis strain AI06, an endophyte of the amazonian açaí palm. Genome announcements, 2(6), e01225-14.

Abe Sato, S. T., Marques, J. M., da Luz de Freitas, A., Sanches Progenio, R. C., Nunes, M. R. T., Mota de Vasconcelos Massafra, J., ... & Rogez, H. (2021). Isolation and genetic identification of endophytic lactic acid bacteria from the Amazonian açai fruits: Probiotics features of selected strains and their potential to inhibit pathogens. Frontiers in microbiology, 11, 610524.

Peters, L. P., Prado, L. S., Silva, F. I., Souza, F. S., & Carvalho, C. M. (2020). Selection of endophytes as antagonists of Colletotrichum gloeosporioides in açaí palm. Biological Control, 150, 104350.

Batista, B. N., Matias, R. R., & Albuquerque, P. M. (2022). Hydrolytic enzyme production from açai palm (Euterpe precatoria) endophytic fungi and characterization of the amylolytic and cellulolytic extracts. World Journal of Microbiology and Biotechnology, 38(2), 1-13.

Or the older work of Rodrigues:

Rodrigues, K. F. (1992). Endophytic fungi in the tropical palm Euterpe oleracea Mart. City University of New York.

Rodrigues, K. F. (1994). The foliar fungal endophytes of the Amazonian palm Euterpe oleracea. Mycologia, 86(3), 376-385.

However, older works are sometimes no longer perceived as a relevant source, as they are based only on morphological identification.

L83-L84: This is the biggest shortage of work. The determination of the fungi should have been done on the basis of molecular methods that are already available today, and not on the basis of literature that is at least twenty years old (two sources are even from 1959 and 1971). In the list of "species" in Table 2, you include e.g. Aspergillus sp 26. It is clear that it does not have to be a specific species of Aspergillus - on the one hand, it does not have to be Aspergillus at all, on the other hand, it can be a strain, not the species. Individual strains may be of interest to other scientists, and if they isolate them from their sample and determine them molecularly (as is the standard today), it would be a benefit for science to be able to associate your ecological knowledge with this strain, which will not be possible in the current form of your results. What you labeled "Aspergillus sp 26" will be difficult for other scientists to replicate.

Further comments:

L59: „this article aims to isolated“… No, the purpose of your research was to isolate, and the purpose of your paper is to inform about that research. It is not directly the goal of the article to isolate, the article cannot do that.

L69: How many samples did you receive, from how many samples were fungi isolated? It would be appropriate to add the species accumulation curve, which would show us how the number of isolated fungi morphotypes grew with the number of samples, so that we would have an idea if this curve reaches an asymptote, or if it would be necessary to sample more.

L85-L86: The list of species should probably already be part of the results. On the contrary, L169 should be part of the methodology.

L241: Why is there a comparison with an orchid? And if this is the comparison with endophytes from other plants growing in the given environment, why aren't there more citations?

L501: This is not an appropriate reference format

Numerous errors in grammar or formatting:

L72: „august“ should be August

L76: Different font

L82: „it was mate use of an optical“… very strange formulation

L93, L119, L21, L125 and in many other places. Sentence starts with lower case.

L118 and elsewhere: „Pestalotiopsis sp e Aspergillus sp“ – „sp“ should be „sp.“, „e“ is not translated

L192: „The isolated“ – the isolates?

L248: „toll“ – tool?

L255: „93, 87 %“ – 93.87%?

L256, L261 and perhaps elsewhere: „Aspergillus“ should be in italics

L279: „When using the endophytic fungus Aspergillus sp, the factors temperature (L), pH (Q).“ … are significant? Part of the sentence is evidently missing.

L284-L285: This sentence is also grammatically strange

L361: The second part of the species name should be in lower case

L386: „The pH ha a vital role“ – has?

L389: „biothecnological“

L406-L407: This sentence and the strange ordering of quotations 39 to 41 are difficult to understand

L412: „genus“ should be „genera“

L412: „Whom“?

Author Response

Reviewer 2:

L59: „this article aims to isolated“… No, the purpose of your research was to isolate, and the purpose of your paper is to inform about that research. It is not directly the goal of the article to isolate, the article cannot do that.

  1. Text corrected as suggested.

L69: How many samples did you receive, from how many samples were fungi isolated? It would be appropriate to add the species accumulation curve, which would show us how the number of isolated fungi morphotypes grew with the number of samples, so that we would have an idea if this curve reaches an asymptote, or if it would be necessary to sample more.

R = Não foi realizado o quantitativo de isolados por parte do vegetal utilizado.

L85-L86: The list of species should probably already be part of the results. On the contrary, L169 should be part of the methodology.

  1. The list of isolates was kept in the methodology to facilitate the reader's understanding, regarding the following methodological steps that were performed with only 2 isolated strains.

L241: Why is there a comparison with an orchid? And if this is the comparison with endophytes from other plants growing in the given environment, why aren't there more citations?

  1. The text was corrected and new refers added.

L501: This is not an appropriate reference format

Numerous errors in grammar or formatting:

L72: „august“ should be August

  1. Text corrected.

L76: Different font

  1. Text corrected.

L82: „it was mate use of an optical“… very strange formulation

  1. Text corrected.

L93, L119, L21, L125 and in many other places. Sentence starts with lower case.

  1. Text corrected.

L118 and elsewhere: „Pestalotiopsis sp e Aspergillus sp“ – „sp“ should be „sp.“, „e“ is not translated

  1. Text corrected.

L192: „The isolated“ – the isolates?

  1. Text corrected.

L248: „toll“ – tool?

  1. Text corrected.

L255: „93, 87 %“ – 93.87%?

  1. Text corrected.

L256, L261 and perhaps elsewhere: „Aspergillus“ should be in italics

  1. Text corrected.

L279: „When using the endophytic fungus Aspergillus sp, the factors temperature (L), pH (Q).“ … are significant? Part of the sentence is evidently missing.

  1. Text corrected.

L361: The second part of the species name should be in lower case

  1. Text corrected.

L386: „The pH ha a vital role“ – has?

  1. Text corrected.

L389: „biothecnological“

  1. Text corrected.

L406-L407: This sentence and the strange ordering of quotations 39 to 41 are difficult to understand

L412: „genus“ should be „genera“

  1. Text corrected.

L412: „Whom“?

  1. Text corrected.

Reviewer 3 Report

This manuscript describes in vitro activities of isolated endophytic fungi from of fruit of Euterpe oleracea Mart (Açaizeiro), and demonstrates their potential enzymatic from lipolytic activity.  The manuscript was poorly written and it needs to be reviewed carefully.

Comments:

-         Title needs to rewrite to become more attractive and showed the content of the article.

-         Abstract needs to contain some numerical results.

-         Introduction very short: state more about Brazilian Amazon plants, Euterpe oleracea Mart (Açaizeiro),  endophytic fungi  associated with Euterpe oleracea, and lipase enzymatic activity by endophytic fungi  isolated from Euterpe oleracea Mart (Açaizeiro) from the Brazilian Amazon.

-         Identification and conservation of endophytic fungi by morphological characters is not enough. If the authors have more confirmation please add it. Also if you have the phylogenetic characterization of these isolates, please add it to the article.

-         The conclusion is too general, it should be connected and supported by the results.

-         please revise all names of fungi  in the text and change them to italic

-         - Abstract and Conclusion section needs improvement. Please highlight the important findings of the manuscript

-          

Author Response

Reviewer 3:

1) Title needs to rewrite to become more attractive and showed the content of the article.

  1. The title was altered for: Euterpe oleracea Mart (Açaizeiro) from the Brazilian Amazon: A novel font of fungi for lipase production.

2) Abstract needs to contain some numerical results.

  1. The abstract was corrected.

3) Introduction very short: state more about Brazilian Amazon plants, Euterpe oleracea Mart (Açaizeiro),  endophytic fungi  associated with Euterpe oleracea, and lipase enzymatic activity by endophytic fungi  isolated from Euterpe oleracea Mart (Açaizeiro) from the Brazilian Amazon.

  1. News references were added.

4)  Identification and conservation of endophytic fungi by morphological characters is not enough. If the authors have more confirmation please add it. Also if you have the phylogenetic characterization of these isolates, please add it to the article.

  1. Pela escassez de financiamento não foi possível identificar as espécies dos fungos isolados. No entanto, trata-se de um trabalho promissor que certamente terá continuidade de investigação.

5) Please revise all names of fungi in the text and change them to italic.

  1. Text revised and corrected.

6) Abstract and Conclusion section needs improvement. Please highlight the important findings of the manuscript

  1. Text revised and corrected.

Round 2

Reviewer 2 Report

I am not satisfied with the authors response. The authors responded to my minor comments, which could have been corrected in a few minutes, and chose to completely ignore the responses to the more serious criticisms. It is understandable that they do not fulfill everything that the reviewer wants, but at least the answer would be appropriate. That's why I'm copying the unanswered comments into a new answer to the authors.

The most significant criticisms

L41: „However, studies about endophytic fungi from E. oleracea are limited in literature or scare.“ They are not so rare, it is possible to mention these working with E. oleracea or a relative E. precatoria:

McCulloch, J. A., de Oliveira, V. M., de Almeida Pina, A. V., Pérez-Chaparro, P. J., de Almeida, L. M., de Vasconcelos, J. M., ... & Nunes, M. R. T. (2014). Complete genome sequence of Lactococcus lactis strain AI06, an endophyte of the amazonian açaí palm. Genome announcements, 2(6), e01225-14.

Abe Sato, S. T., Marques, J. M., da Luz de Freitas, A., Sanches Progenio, R. C., Nunes, M. R. T., Mota de Vasconcelos Massafra, J., ... & Rogez, H. (2021). Isolation and genetic identification of endophytic lactic acid bacteria from the Amazonian açai fruits: Probiotics features of selected strains and their potential to inhibit pathogens. Frontiers in microbiology, 11, 610524.

Peters, L. P., Prado, L. S., Silva, F. I., Souza, F. S., & Carvalho, C. M. (2020). Selection of endophytes as antagonists of Colletotrichum gloeosporioides in açaí palm. Biological Control, 150, 104350.

Batista, B. N., Matias, R. R., & Albuquerque, P. M. (2022). Hydrolytic enzyme production from açai palm (Euterpe precatoria) endophytic fungi and characterization of the amylolytic and cellulolytic extracts. World Journal of Microbiology and Biotechnology, 38(2), 1-13.

Or the older work of Rodrigues:

Rodrigues, K. F. (1992). Endophytic fungi in the tropical palm Euterpe oleracea Mart. City University of New York.

Rodrigues, K. F. (1994). The foliar fungal endophytes of the Amazonian palm Euterpe oleracea. Mycologia, 86(3), 376-385.

However, older works are sometimes no longer perceived as a relevant source, as they are based only on morphological identification.

L83-L84: This is the biggest shortage of work. The determination of the fungi should have been done on the basis of molecular methods that are already available today, and not on the basis of literature that is at least twenty years old (two sources are even from 1959 and 1971). In the list of "species" in Table 2, you include e.g. Aspergillus sp 26. It is clear that it does not have to be a specific species of Aspergillus - on the one hand, it does not have to be Aspergillus at all, on the other hand, it can be a strain, not the species. Individual strains may be of interest to other scientists, and if they isolate them from their sample and determine them molecularly (as is the standard today), it would be a benefit for science to be able to associate your ecological knowledge with this strain, which will not be possible in the current form of your results. What you labeled "Aspergillus sp 26" will be difficult for other scientists to replicate.

L69: How many samples did you receive, from how many samples were fungi isolated? It would be appropriate to add the species accumulation curve, which would show us how the number of isolated fungi morphotypes grew with the number of samples, so that we would have an idea if this curve reaches an asymptote, or if it would be necessary to sample more.... I am not satisfied with your answer.

Author Response

Comments and Suggestions for Authors

Reviewer 1:

Pont 1. I have one observation to made: at the reference number 39, the name of species Phoenix Dactylifera L. is not italic. Please to correct!

R = The text was corrected.

Pont 2. The manuscript contains a sufficient number of citations, although some of them are already outdated and I am missing some newer ones. The statistical analysis of the data is well done, but I have concerns about the input data, which should have been obtained on the basis of molecular identification of fungi and not just morphological, moreover, based on 60-year-old literature. This only morphological determination casts doubt on the authors' results, but can be easily remedied by additional species determination. 

L41: “However, studies about endophytic fungi from E. oleracea are limited in literature or scare”. They are not so rare, it is possible to mention these working with E. oleracea or a relative E. precatoria:

  1. Refers all suggested were added in text, such as:

The first description of endophytic fungi in E. oleracea palm leaves, from the Amazon region, was performed by Rodrigues [13 ]. Rodrigues also described the occurrence of a new genus, Letendraeopsis palmarum, and new species of the genus Idrella isolated from E. oleracea palm [ref ]. The enzymatic potential of endophytic fungi isolated from açai palms (Euterpe precatoria Mart.) were show by Batista [13], this study cellulolytic and amylolytic extracts showed the highest enzymatic activities. Recently, the antimicrobial extract of endophytic fungi of E. precatoria against Staphylococcus aureus, Streptococcus pneumoniae, Enterococcus faecalis, Escherichia coli and Klebsiella pneumoniae human pathogens was examined [14], in addition endophytic fungi from E. precatoria were used as antagonistic agents towards Colletotrichum gloeosporioides in the control of anthracnose in açaí leaflets [15]. McCulloch et al. [ref ] reported the genome, in a single chromosome, of Lactococcus lactis strain AI06, isolated from the mesocarp of the açaí fruit (Euterpe oleracea) in eastern Amazonia, Brazil. This strain is an endophyte of the açaí palm and also a component of the microbiota of the edible food product.However, studies about endophytic fungi from E. oleracea are scare in literature. [13] Rodrigues, K. F. (1992). Endophytic fungi in the tropical palm Euterpe oleracea Mart. City University of New York.

[14] Rodrigues, K. F. (1994). The foliar fungal endophytes of the Amazonian palm Euterpe oleracea. Mycologia, 86(3), 376-385.

[15] Batista, B. N., Matias, R. R., & Albuquerque, P. M. (2022). Hydrolytic enzyme production from açai palm (Euterpe precatoria) endophytic fungi and characterization of the amylolytic and cellulolytic extracts. World Journal of Microbiology and Biotechnology, 38(2), 1-13.

[16] Batista, B.N.; Raposo, N.V. de M.; Silva, I.R. da Isolamento e Avaliação Da Atividade Antimicrobiana de Fungos Endofíticos de Açaizeiro. Rev. Fitos 2018, 12, 161–174, doi:10.5935/2446-4775.20180015.

[17] Peters, L. P., Prado, L. S., Silva, F. I., Souza, F. S., & Carvalho, C. M. (2020). Selection of endophytes as antagonists of Colletotrichum gloeosporioides in açaí palm. Biological Control, 150, 104350.

[18] McCulloch, J. A., de Oliveira, V. M., de Almeida Pina, A. V., Pérez-Chaparro, P. J., de Almeida, L. M., de Vasconcelos, J. M., ... & Nunes, M. R. T. (2014). Complete genome sequence of Lactococcus lactis strain AI06, an endophyte of the amazonian açaí palm. Genome announcements, 2(6), e01225-14.

The refer following not was cited. Because the focus of this work was endophytic bacteria and we understand that it is beyond the scope of the subject addressed.

Abe Sato, S. T., Marques, J. M., da Luz de Freitas, A., Sanches Progenio, R. C., Nunes, M. R. T., Mota de Vasconcelos Massafra, J., ... & Rogez, H. (2021). Isolation and genetic identification of endophytic lactic acid bacteria from the Amazonian açai fruits: Probiotics features of selected strains and their potential to inhibit pathogens. Frontiers in microbiology, 11, 610524.

L83-L84: This is the biggest shortage of work. The determination of the fungi should have been done on the basis of molecular methods that are already available today, and not on the basis of literature that is at least twenty years old (two sources are even from 1959 and 1971). In the list of "species" in Table 2, you include e.g. Aspergillus sp 26. It is clear that it does not have to be a specific species of Aspergillus - on the one hand, it does not have to be Aspergillus at all, on the other hand, it can be a strain, not the species. Individual strains may be of interest to other scientists, and if they isolate them from their sample and determine them molecularly (as is the standard today), it would be a benefit for science to be able to associate your ecological knowledge with this strain, which will not be possible in the current form of your results. What you labeled "Aspergillus sp 26" will be difficult for other scientists to replicate.

  1. Concord as a reviewer on the importance of molecular identification to confirm genus and species. However, for this work it was not possible, however, for future work, some strains will certainly be treated for their species. All the isolates used in the present work were characterized based on the macroscopic characteristics of the colonies, as well as the microscopic characteristics of the conidiophores and conidia accessed in specialized literature.

Reviewer 2:

L59: „this article aims to isolated“… No, the purpose of your research was to isolate, and the purpose of your paper is to inform about that research. It is not directly the goal of the article to isolate, the article cannot do that.

  1. Text corrected as suggested.

L69: How many samples did you receive, from how many samples were fungi isolated? It would be appropriate to add the species accumulation curve, which would show us how the number of isolated fungi morphotypes grew with the number of samples, so that we would have an idea if this curve reaches an asymptote, or if it would be necessary to sample more.

R = Não foi realizado o quantitativo de isolados por parte do vegetal utilizado.

L85-L86: The list of species should probably already be part of the results. On the contrary, L169 should be part of the methodology.

  1. The list of isolates was kept in the methodology to facilitate the reader's understanding, regarding the following methodological steps that were performed with only 2 isolated strains.

L241: Why is there a comparison with an orchid? And if this is the comparison with endophytes from other plants growing in the given environment, why aren't there more citations?

  1. The text was corrected and new refers added.

L501: This is not an appropriate reference format

Numerous errors in grammar or formatting:

L72: „august“ should be August

  1. Text corrected.

L76: Different font

  1. Text corrected.

L82: „it was mate use of an optical“… very strange formulation

  1. Text corrected.

L93, L119, L21, L125 and in many other places. Sentence starts with lower case.

  1. Text corrected.

L118 and elsewhere: „Pestalotiopsis sp e Aspergillus sp“ – „sp“ should be „sp.“, „e“ is not translated

  1. Text corrected.

L192: „The isolated“ – the isolates?

  1. Text corrected.

L248: „toll“ – tool?

  1. Text corrected.

L255: „93, 87 %“ – 93.87%?

  1. Text corrected.

L256, L261 and perhaps elsewhere: „Aspergillus“ should be in italics

  1. Text corrected.

L279: „When using the endophytic fungus Aspergillus sp, the factors temperature (L), pH (Q).“ … are significant? Part of the sentence is evidently missing.

  1. Text corrected.

L361: The second part of the species name should be in lower case

  1. Text corrected.

L386: „The pH ha a vital role“ – has?

  1. Text corrected.

L389: „biothecnological“

  1. Text corrected.

L406-L407: This sentence and the strange ordering of quotations 39 to 41 are difficult to understand

L412: „genus“ should be „genera“

  1. Text corrected.

L412: „Whom“?

  1. Text corrected.

Reviewer 3:

1) Title needs to rewrite to become more attractive and showed the content of the article.

  1. The title was altered for: Euterpe oleracea Mart (Açaizeiro) from the Brazilian Amazon: A novel font of fungi for lipase production.

2) Abstract needs to contain some numerical results.

  1. The abstract was corrected.

3) Introduction very short: state more about Brazilian Amazon plants, Euterpe oleracea Mart (Açaizeiro),  endophytic fungi  associated with Euterpe oleracea, and lipase enzymatic activity by endophytic fungi  isolated from Euterpe oleracea Mart (Açaizeiro) from the Brazilian Amazon.

  1. News references were added.

4)  Identification and conservation of endophytic fungi by morphological characters is not enough. If the authors have more confirmation please add it. Also if you have the phylogenetic characterization of these isolates, please add it to the article.

  1. Pela escassez de financiamento não foi possível identificar as espécies dos fungos isolados. No entanto, trata-se de um trabalho promissor que certamente terá continuidade de investigação.

5) Please revise all names of fungi in the text and change them to italic.

  1. Text revised and corrected.

6) Abstract and Conclusion section needs improvement. Please highlight the important findings of the manuscript

  1. Text revised and corrected.

Prof. Dr. Irlon M. Ferreira

Universidade Federal do Amapá

Grupo de Biocatálise e Biotransformação em Química Orgânica
Rod. Juscelino Kubitschek, Km 02, Jardim Marco Zero, Macapá-AP, Brasil, CEP.: 68902-280
Currículo Lattes http://lattes.cnpq.br/9897023410899133

Reviewer 3 Report

Accept in present form

Author Response

(The authors gave the same response as above.)
